# Can Finite Element Method Obtain SVET Current Densities Closer to True Localized Corrosion Rates?

**DOI:** 10.3390/ma15113764

**Published:** 2022-05-24

**Authors:** Mohsen Saeedikhani, Sareh Vafakhah, Daniel J. Blackwood

**Affiliations:** 1Department of Materials Science and Engineering, National University of Singapore, 9 Engineering Drive 1, Singapore 117576, Singapore; msemohs@nus.edu.sg; 2School of Chemical and Biomolecular Engineering, The University of Sydney, Darlington, NSW 2006, Australia; sareh.vafakhah@sydney.edu.au

**Keywords:** SVET, conductivity, diffusion, current density, FEM

## Abstract

In this paper, the finite element method was used to simulate the response of the scanning vibrating electrode technique (SVET) across an iron–zinc cut-edge sample in order to provide a deeper understanding of the localized corrosion rates measured using SVET. It was found that, if the diffusion layer was neglected, the simulated current density using the Laplace equation fitted the experimental SVET current density perfectly. However, the electrolyte was not perturbed by a vibrating SVET probe in the field, so a diffusion layer existed. Therefore, the SVET current densities obtained from the local conductivity of the electrolyte would likely be more representative of the true corrosion rates than the SVET current densities obtained from the bulk conductivity. To help overcome this difference between natural conditions and those imposed by the SVET experiment, a local electrolyte corrected conductivity SVET (LECC-SVET) current density was introduced, which was obtained by replacing the bulk electrolyte conductivity measured experimentally by the local electrolyte conductivity simulated using the Nernst−Einstein equation. Although the LECC-SVET current density did not fit the experimental SVET current density as perfectly as that obtained from the Laplace equation, it likely represents current densities closer to the true, unperturbed corrosion conditions than the SVET data from the bulk conductivity.

## 1. Introduction

The vibrating probe technique was initially developed for biological studies [1,2]. Thanks to the pioneering work of Isaacs, the scanning vibrating probe technique (SVET) was adapted to corrosion studies [3,4,5,6] and SVET is now being used extensively to investigate a wide range of corrosion phenomena, as summarized briefly in a review paper by Bastos et al. [7].

The SVET probe is typically held 100–200 µm above the sample’s surface and oscillates a few tens of micrometers perpendicular to the sample’s surface (occasionally, the oscillation may be set to be parallel to the sample’s surface directions depending on the instrument model/setup) to collect the potential gradient in the electrolyte. The potential gradient is then converted to the solution current density by Ohm’s law:(1)$$j^{\mathrm{SVET}}=-{\;\sigma }^{\mathrm{bulk}}\;\frac{\Delta V}{\Delta r}$$
where $j^{\mathrm{SVET}}\;\left(\frac{A}{m^{2}}\right)$ is the electrolyte local current density, $\sigma^{\mathrm{bulk}}\left(\frac{S}{m}\right)$ is the bulk solution conductivity, ΔV (V) is the electrolyte’s potential gradient, and Δr is the oscillation distance of the probe tip. An SVET map is obtained by the probe scanning parallel to the sample’s surface.

Not only is SVET useful for studying local electrochemical phenomena, but it is also often a prime choice for the verification of simulated corrosion current densities [8,9,10,11]. Researchers depict how closely their simulated current density fits the SVET current density. On this basis, the closer the fit, the more accurate the simulation. However, most often, a perfect fit is not obtained between SVET and simulated current densities, owing to the following: (1) the simplicity of the model compared with the complexity of the corrosion process, or (2) fundamental differences between the physics of the SVET and the simulation. The second case does not necessarily mean that the simulation is not accurate or needs modification. In fact, on the contrary, sometimes simpler models fit the SVET better [8,12,13], which will be discussed here.

First, it is necessary to briefly explain what is meant by “simple” models. In general, there are two main approaches to the finite element method in corrosion studies—the Laplacian approach and the Nernst−Planck (NP) approach, with detailed explanations published elsewhere [14,15]. In the Laplacian approach, the Laplace equation ($\nabla^{2}\varphi_{l}=0$) will be solved over the electrolyte’s domain with uniform resistance. However, when the electrolyte resistance is not uniform, the Laplacian approach is not the most accurate. This Nernst−Planck approach solves the NP equation (governing equations will be discussed in Section 3) over the electrolyte domain to simulate the transport of species by diffusion, migration, and convection; in stagnant electrolytes, the convection term is neglected. The NP approach is considered a complete means of solving transient problems in corrosion studies, as it solves for the concentration gradient, electrolyte current density, and electrolyte potential distribution. A drawback of the NP approach is that it is computationally expensive because of the consideration of many parameters that need to be solved. Here, “simple” models are regarded as those that have used the Laplacian approach alone or in conjunction with Fick’s laws of diffusion.

Thébault et al. extensively studied the simulation of iron–zinc cut-edge corrosion. They started with a potential model (named electrostatic model) [8] that solves the Laplace equation in the electrolyte domain, which is comprised of only NaCl and oxygen. They continued to develop their model by incorporating multiple species and homogeneous reactions in the electrolyte, naming it the coupled electrochemical transport reaction model (CETR model) [9]. Their CETR model aimed to take into account the contribution of charge carriers in the local current density. Finally, they showed that their electrostatic model, even as a simple model, fitted the experimental SVET data fairly well and the CETR model perfectly fitted the SVET current density only without the consideration of species diffusion [12]. Surprisingly, Thébault et al. [12] found that adding a diffusion layer to their CETR model resulted in a poorer fit to the SVET current density, even though, theoretically, it was a more rigorous model. They believed that convection (stirring) due to the SVET probe oscillations locally annihilated the diffusion layer, as if the SVET measurements were performed in a homogeneous medium (Figure 1a).

Through simulation and experimentation studies, Dolgikh et al. [16] concluded that local mixing leads to a substantial increase in the migration current density in the vicinity of the probe, with a simultaneous decrease in the diffusion current contribution (see the NP equation in Section 3). They concluded that, even though SVET is based on the measurement of the ohmic current density, it always measures the total current density because of the local mixing caused by the probe vibration. Recently, Charles-Granville et al. [13] developed an FEM model using the Laplacian approach in conjunction with Fick’s second law to study the macro-galvanic corrosion of AA7050 and SS316 alloys. They modified the boundary conditions in such a way that their “simple” model fitted reasonably well to the SVET data, despite the fact that they had studied alloys comprised of multiple elements rather than only a single element such as zinc or iron [17,18].

Such studies may justify using the bulk electrolyte’s conductivity in Equation (1) to obtain the SVET current density from the SVET potential gradient. However, in true (natural) corrosion phenomena, the environment is not perturbed by a vibrating probe and the diffusion layer is not annihilated (Figure 1b). The authors of this paper previously demonstrated the difference between the SVET current density in the electrolyte and the current density on the metal surface [19]. Here, the focus will be to use “local conductivity” instead of “bulk conductivity” to obtain SVET current densities closer to the true corrosion rates. The “local conductivity” is obtained through the simulation at an average height where the SVET probe oscillates.

Despite the fact that SVET has often been used for the verification of simulated current densities, this paper, on the contrary, studies the feasibility of using FEM to obtain SVET local current densities closer to the real corrosion phenomena. As such, the local conductivity along the cut-edge surface of a Fe–Zn couple was extracted from an accurate multi-ion transport and reaction model (MITReM, which is the same concept as CETR) change [20,21] and fed to Ohm’s law (Equation (1)) to obtain the local current densities that would occur in the electrolyte in the presence of a diffusion layer; that is, the actual corrosion system unperturbed by the SVET probe’s oscillations.

Finally, it is essential to note that SVET is still a valuable technique, which is likely irreplaceable by models. SVET has some advantages over other electrochemical techniques [22,23]; SVET clearly visualizes the anodic and cathodic regions, whereas corrosion models still have to consider the anodic and cathodic areas as a “priori” (pre-defined polarities for the electrodes) [15]. However, SVET in conjunction with other models will likely produce more accurate data and provide further insight into the underlying corrosion mechanisms [19].

## 2. Materials and Methods

### 2.1. Samples

A zinc coating (10 µm thickness) was electroplated on one side of the iron samples (500 µm cross section thickness) by a commercial company. The samples were cut to 20 mm × 20 mm and mounted vertically in a room-temperature-cure epoxy system so that only one cross section (cut-edge) of the sample was exposed to the electrolyte. The cut-edge side was ground with SiC paper up to grade 4000, polished with polishing cloths from Struers, and rinsed with ethanol. Cut-edge samples were used to avoid issues related to the variations in the probe to sample height over the anode and cathodes that occur with scratched zinc coatings [19]. Figure 2 shows a schematic of the samples. The samples were observed by optical microscopy to ensure the zinc coating was not damaged on the cut-edge side.

### 2.2. SVET

The SVET measurements (Uniscan Instrument M370, Princeton Applied Research) were performed with a 10 µm diameter platinized platinum microprobe tip controlled by the dedicated software M370. The SVET measurements were taken under a free corrosion potential in a 3 mm thick 3.5 wt% NaCl electrolyte. The electrolyte’s thickness was monitored using a lateral camera and the initial bulk conductivity was 1.13 S·m^−1^. The electrolyte thickness was measured using the probe tip as a guide. The probe tip was first adjusted to be tangent to the zinc surface and was then raised to 3000 μm. Then, the electrolyte was poured into the cell until it was leveled with the probe tip level, which was monitored using the camera.

Moreover, the SVET measurements were taken perpendicular to the cut-edge plane (Z-direction). The microprobe vibrated at approximately 150 µm above the specimens with a 30 µm peak-to-peak amplitude. The length of the line scan was 2500 µm (Figure 2), and comprised of scans over the insulator and the sample. The scan started from the outermost edge of the insulator on one edge, cut through the middle of the cross section, and stopped just before the outermost edge of the insulator on the other edge, with measurement steps of 50 µm. As such, there were a total of 51 measurement points in the line scan. At each point, the probe waited for 0.5 s and then recorded the average current density flowing during the next 0.2 s. The line scan duration was less than 2 min. The current density values were positive and negative for the anodic and cathodic current densities, respectively.

## 3. Simulation

### 3.1. Geometry

Figure 3 shows the two-dimensional geometry of the model and its discretization used in this simulation study. The galvanic cell was a cut-edge sample comprised of iron and zinc, which were in direct physical contact and confined by two lengths of inert insulators. The whole system was immersed in a 3000 µm layer of an aqueous solution comprised of the species listed in Table 1. Two different types of electrolyte domains were considered, as follows: Figure 3a shows an electrolyte that is split into two subdomains: a diffusion layer where the concentration gradients occur (see Figure 1b), hence with a much finer mesh size, and a bulk electrolyte with a coarser mesh size, where the concentration of the species is assumed to remain the same as their initial concentrations, $c_{i}^{0}$
(Table 1); Figure 3b shows a single electrolyte domain without a diffusion layer, such that there are no concentration gradients.

### 3.2. Theory and Governing Equations

The non-convective Nernst−Planck equation allows for calculating the ionic current caused by the transport of charge carriers in the electrolyte:(2)$$J_{i}=-{\;D}_{i}\nabla c_{i}-z_{i}\frac{D_{i}}{\mathrm{RT}}{\mathrm{Fc}}_{i}\nabla \emptyset$$
where $D_{i}$ is the diffusion coefficient, $c_{i}$ is the concentration, $z_{i}$ is the charge, R is the gas constant, T is the temperature, F is the Faraday constant, and ∇∅ is the potential gradient in the electrolyte. The electrolyte considered was an aqueous solution (taking the water dissociation reaction into account) containing 3.5 wt% NaCl initially at pH 6.5. Moreover, 12 species were considered in the electrolyte and their initial concentrations ($c_{i}^{0}$) and diffusion coefficients are reported in Table 1. All of the species were considered soluble and their precipitation in a form that might cause electrode surface blockage was not considered in this stationary simulation study. This is in line with the SVET measurements taken immediately after immersion, before the solid corrosion products ever precipitated.

The mass balance for the charge carriers in the electrolyte was given by the relationship below:(3)$$\frac{\partial c_{i}}{\partial t}=-\nabla J_{i}+R_{i}$$
where $J_{i}$ is the ionic current from Equation (2) and $R_{i}$ is the production term of species i. Table 2 reports the homogeneous reactions considered in this study along with their equilibrium constants.

Finally, the bulk electrolyte was assumed to be electroneutral:(4)$$\overset{n}{\underset{i=1}{\sum }}z_{i}c_{i}=c_{OH^{-}}-c_{H^{+}}$$

The right-hand side of Equation (4) is as a result of the water dissociation reaction, which was separated from the term $z_{i}c_{i}$.

### 3.3. Boundary Conditions

Oxygen reduction on iron as well as zinc oxidation were considered to be the two main electrochemical cathodic and anodic reactions, respectively:(5)$$j_{O_{2}}=-4Fk_{O_{2}}c_{O_{2}}\exp\left(-\frac{4F}{RT}\alpha_{O_{2}}\left(E-E_{O_{2}}^{0}\right)\right)$$
(6)$$j_{Zn}=2Fk_{Zn}\exp\left(\frac{2F}{RT}\alpha_{Zn}\left(E-E_{Zn}^{0}\right)\right)$$

$k_{O_{2}}$ and $k_{Zn}$ are the interfacial rate constants; $E_{O_{2}}^{0}$ and $E_{Zn}^{0}$ are the equilibrium potentials; and E is the potential difference between the electrode and the electrolyte near the electrode surface, but outside the double layer. In Equation (5), $c_{O_{2}}$ is the local concentration of oxygen being reduced on the iron, as calculated by the model. Moreover, $\alpha_{O_{2}}$ and $\alpha_{Zn}$ are the transfer coefficients for oxygen reduction reaction and zinc oxidation, respectively. The kinetic parameters in Equations (5) and (6) taken from the literature are reported in Table 1.

The initial concentrations of all species $c_{i}^{0}$ were applied on boundaries (1)–(6) in Figure 2, where applicable. Finally, the concentration gradients of the species were only considered within the diffusion layer (diffusion controlled current, see Figure 1b).

## 4. Results and Discussion

Figure 4 shows the simulated current density without a diffusion layer (schematic in Figure 1a), which perfectly fits the experimental SVET data. As mentioned previously, Thébault et al. [12] explained that this is because the SVET probe’s vibration locally annihilates the concentration gradient, and hence the diffusion layer, so that the SVET current density is akin to the ohmic current density (Equation (1)), i.e., an electrolyte with uniform conductivity.

In contrast, Figure 4 also shows that the simulated current density with a diffusion layer (see Figure 1b) evidently over-estimates the experimental SVET current density. As SVET produces the ohmic current density (Equation (1)) and MITReM produces a current density comprised of ohmic, diffusion, and migration current components (Equation (2)), the important question to consider is, “Which one is more accurate?”. We propose that the method below will likely produce more accurate SVET current density data.

Within the diffusion layer, the (simulated) local conductivity along the surface is given by the Nernst−Einstein equation, as follows:(7)$$\sigma_{x,h}^{loc}=\sum \frac{z_{i}^{2}F^{2}D_{i}c_{i}}{RT}$$
where x,h indicates the local conductivity within the diffusion layer at a certain height h (in Z direction) from the electrodes (which is the same as the average height of the SVET probe in this study, i.e., 150 µm) and along the x direction, i.e., parallel to the electrodes. The parameters on the right-hand side of Equation (7) are the same as in Equation (2). It is evident that $\sigma_{x,h}^{loc}$ is highly dependent on the ion concentrations and their respective charges in the diffusion layer.

Figure 5 depicts the simulated local conductivity of the electrolyte at 150 µm above the electrodes’ surface, with the values being between 1.285 and 1.390 S/m. Evidently, the simulated local conductivity is greater at all x positions than the initial conductivity of 1.13 S/m, especially above the zinc anode and iron cathode, because of the high production of Zn^2+^ and OH^−^, respectively. This indicates that the conductivity of the electrolyte in the diffusion layer of a corroding electrode is higher than the bulk conductivity.

Furthermore, we propose that the replacement of the local conductivity into Equation (1) instead of the bulk conductivity will likely produce SVET current densities (Figure 6) that are more representative of the true corrosion situation. We named the latter current density the local electrolyte corrected conductivity SVET (LECC-SVET) current density, that is:(8)$$j^{LECC-SVET}=-\sigma_{x,h}^{loc}\frac{\Delta V}{\Delta r}=-\sum \frac{z_{i}^{2}F^{2}D_{i}c_{i}}{RT}\frac{\Delta V}{\Delta r}$$

As $\sigma_{x,h}^{loc}$ at all x positions is larger than the bulk electrolyte conductivity, the LECC-SVET current density is therefore larger than the SVET current density (Figure 7). Figure 7 shows that the difference between the SVET and LECC-SVET current density values is more evident directly above the zinc anode and iron cathode, which are the regions where the local conductivity is at its highest difference with the bulk conductivity (Figure 5).

We propose that the LECC-SVET current density is likely to be more representative of real corrosion situations than the SVET current density, as the former takes into account the local conductivity, which is a more realistic case for immersion conditions where diffusion layer(s) and concentration gradients exist over the corroding electrode(s).

## 5. Conclusions

In this study, the simulated current density without a diffusion layer perfectly fits the SVET current density measured across an iron–zinc cut-edge sample, but this is not a true representation of galvanic corrosion, which is not nominally perturbed by a vibrating SVET probe. To help overcome this difference between natural conditions and those imposed by the SVET experiment, FEM was used to obtain the local electrolyte corrected conductivity SVET (LECC-SVET) current density, by replacing the bulk conductivity with the simulated local conductivity. Although the LECC-SVET current density does not fit the experimental SVET current density as perfectly as the Laplacian modeled SVET current density does, we believe that the LECC-SVET current density represents current densities closer to the true corrosion conditions than the SVET data obtained by Ohm’s law.

However, two questions remain. First, which of the MITReM or LECC-SVET current densities are more accurate/realistic? Second, would the potential gradients measured by the SVET be any different if the diffusion layer was present? Hopefully, these answers will be provided by the ongoing studies.

This research was conducted to point out that SVET in conjunction with simulation brings more insights into mechanistic studies of localized corrosion phenomena. SVET is still a valuable technique and is likely irreplaceable by models alone, as the latter still fail to simulate corrosion conditions without pre-setting the polarity of the electrodes (anodic or cathodic). SVET clearly identifies the local anode and cathode regions over corroding surfaces. Lastly, atomistic combined with finite element simulations can give additional mechanistic information [24,25].

## Figures and Tables

**Figure 1 materials-15-03764-f001:**
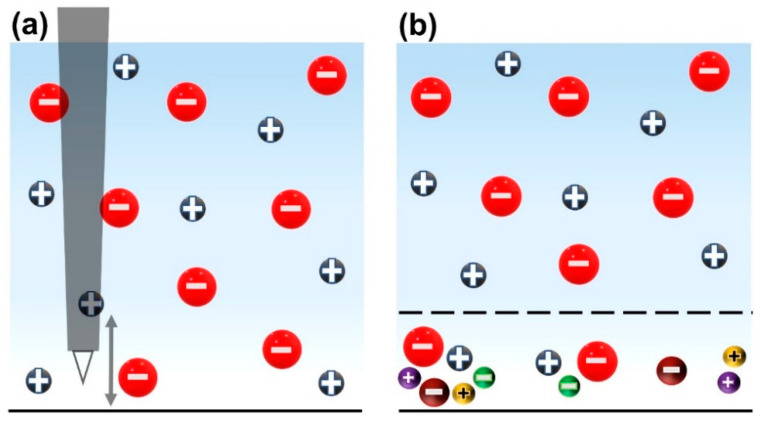
Schematic representation of bulk and diffusion layers over the electrode surface. (**a**) The diffusion layer is annihilated because of the SVET probe vertical oscillation. The arrow depicts the oscillation domain. (**b**) A diffusion layer formed on the corroding electrodes in the natural corrosion cases. The dashed line depicts the transition layer between the diffusion and bulk layers.

**Figure 2 materials-15-03764-f002:**
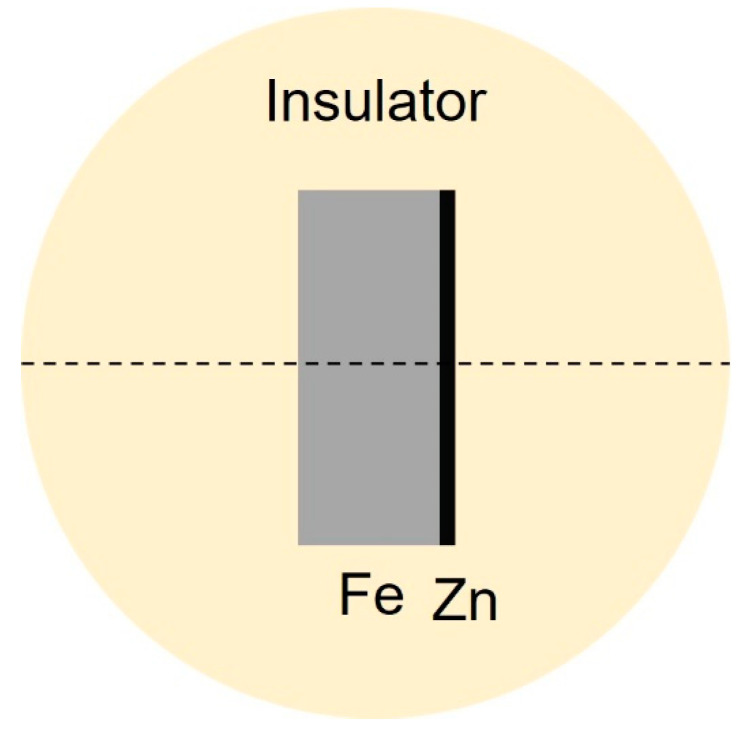
Top view schematic of the cut-edge sample prepared for SVET. The thicknesses of Fe and Zn are 500 µm and 10 µm, respectively. The dashed line depicts the SVET scan line (2500 µm).

**Figure 3 materials-15-03764-f003:**
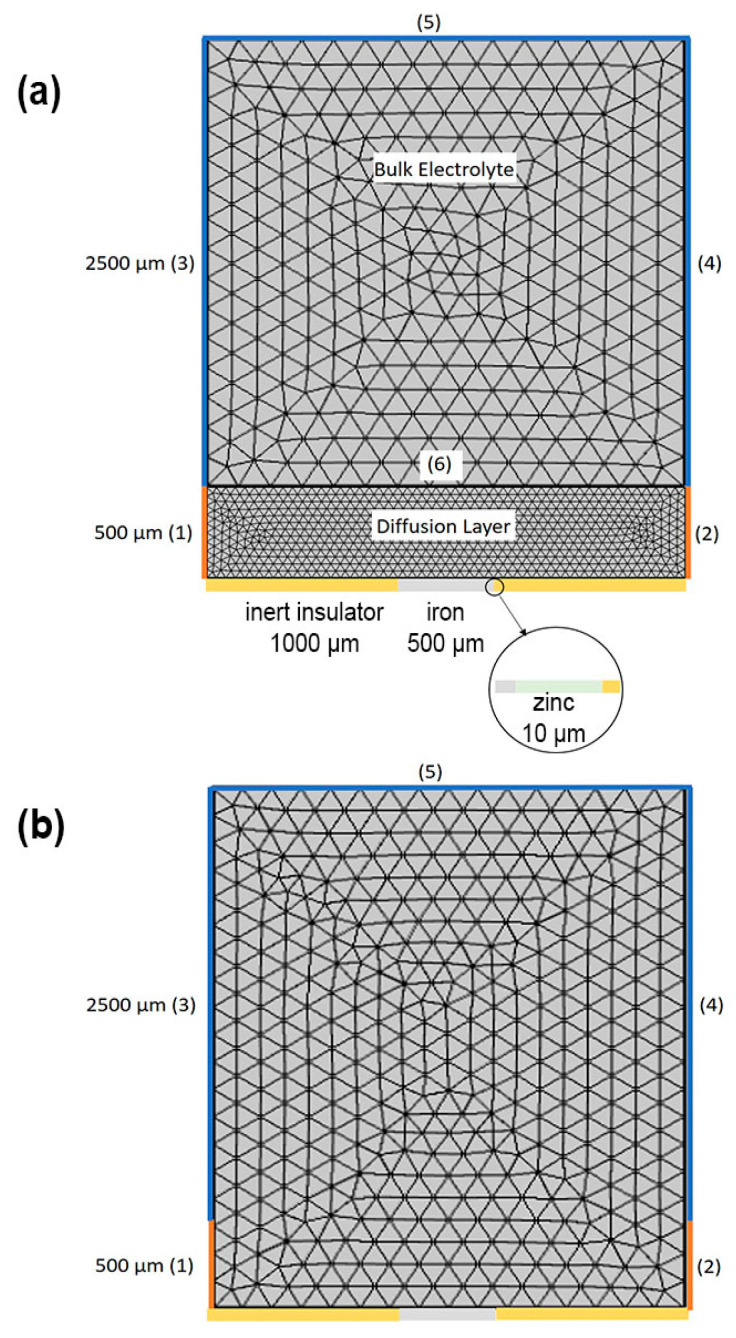
Geometries and meshing used for the simulation (**a**) with a diffusion layer and (**b**) without a diffusion layer. The cut-edge sample is in the middle with the zinc coating on its right-hand side. Lines (1)–(5) depict the boundaries of the electrolyte in diffusion and bulk layers where they exist. Line (6) is the transition from bulk to diffusion layer.

**Figure 4 materials-15-03764-f004:**
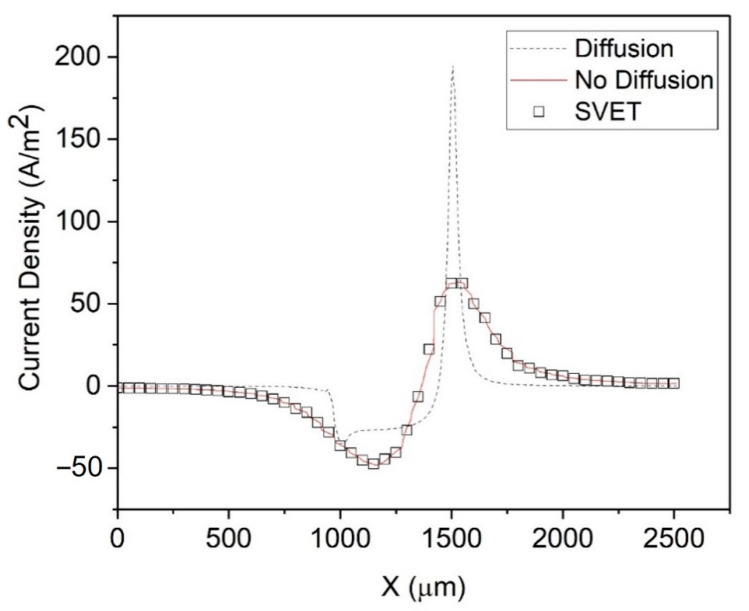
The simulated current density with and without a diffusion layer, along with the SVET current density. Dashed line: simulated current density with a diffusion layer. Solid line: without a diffusion layer. Squares: SVET.

**Figure 5 materials-15-03764-f005:**
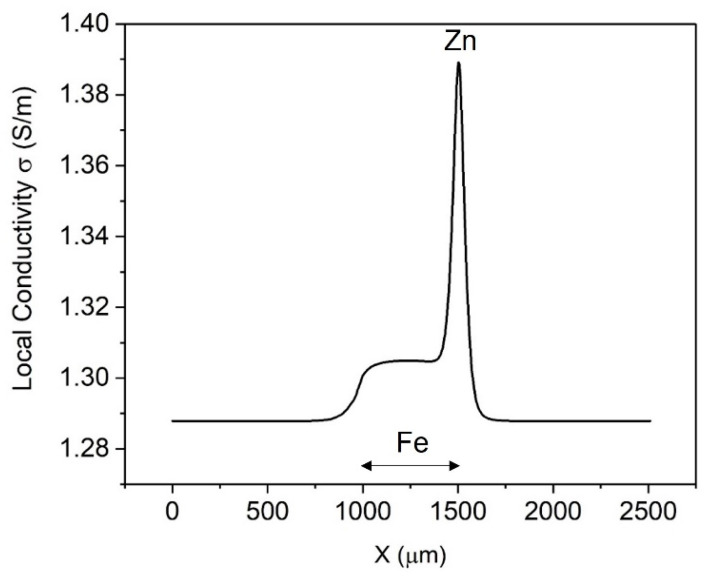
The simulated local conductivity using the Nernst−Einstein equation. The height is 150 µm above the cut-edge sample in the electrolyte with diffusion layer.

**Figure 6 materials-15-03764-f006:**
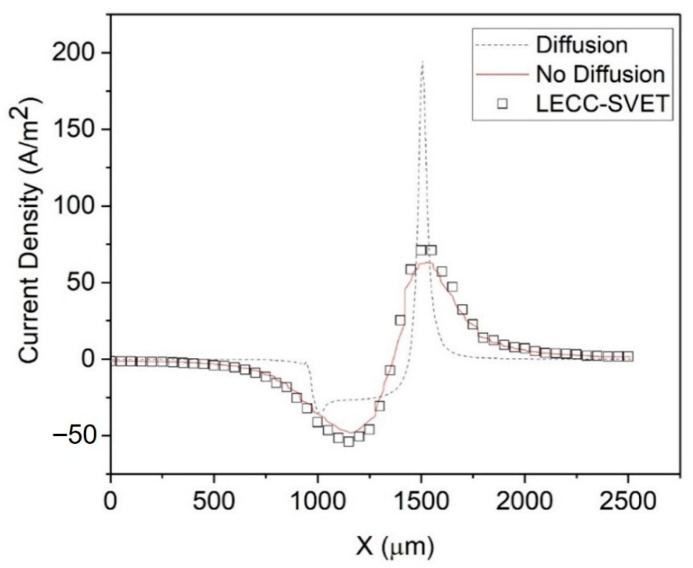
The simulated and SVET current densities. The same as in Figure 4, except the squares show the LECC-SVET current density.

**Figure 7 materials-15-03764-f007:**
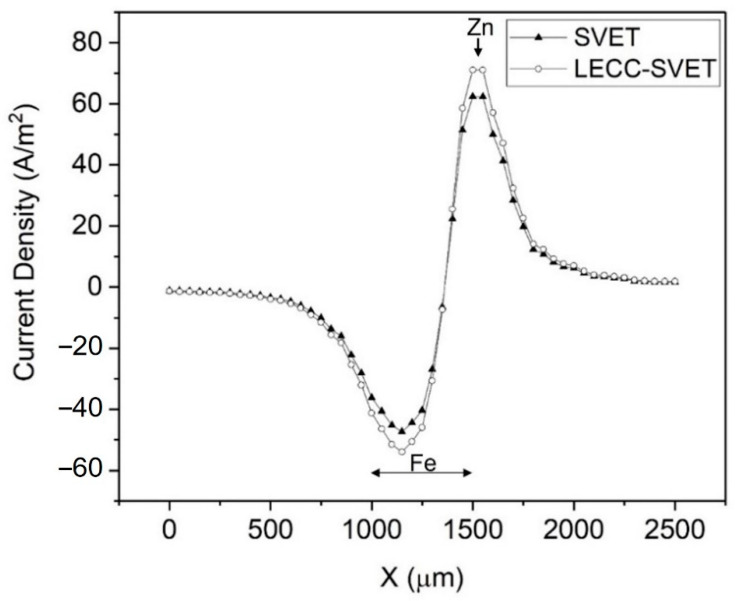
SVET and LECC-SVET current densities. LECC: local electrolyte corrected conductivity.

**Table 1 materials-15-03764-t001:** The constants used in the simulation. $c_{i}^{0}$ is the initial concentration of the species, $D_{i}$ is the diffusion coefficient, $k_{i}$ is the interfacial rate constant, and $\alpha_{i}$ is the transfer coefficient [9,20,21].

Name	Value	Name	Value
$c_{{\mathrm{Na}}^{+}}^{0}$	60 $\mathrm{mol}\cdot m^{-3}$	$D_{{\mathrm{Na}}^{+}}$	1.23 × 10^–9^ $m^{2}\cdot s^{-1}$
$c_{{\mathrm{Cl}}^{-}}^{0}$	60 $\mathrm{mol}\cdot m^{-3}$	$D_{{\mathrm{Cl}}^{-}}$	1.19 × 10^–9^ $m^{2}\cdot s^{-1}$
$c_{O_{2}}^{0}$	0.23 $\mathrm{mol}\cdot m^{-3}$	$D_{O_{2}}$	1.96 × 10^–9^ $m^{2}\cdot s^{-1}$
$c_{{\mathrm{CO}}_{2}}^{0}$	1.31 × 10^–2^ $\mathrm{mol}\cdot m^{-3}$	$D_{{\mathrm{CO}}_{2}}$	1.91 × 10^–9^ $m^{2}\cdot s^{-1}$
$c_{{\mathrm{OH}}^{-}}^{0}$	3 × 10^–8^ $\mathrm{mol}\cdot m^{-3}$	$D_{{\mathrm{OH}}^{-}}$	5.3 × 10^–9^ $m^{2}\cdot s^{-1}$
$c_{H^{+}}^{0}$	3 × 10^–7^ $\mathrm{mol}\cdot m^{-3}$	$D_{H^{+}}$	9.3 × 10^–9^ $m^{2}\cdot s^{-1}$
$c_{{\mathrm{CO}}_{3}^{2-}}^{0}$	3 × 10^–11^ $\mathrm{mol}\cdot m^{-3}$	$D_{{\mathrm{CO}}_{3}{}^{2-}}$	0.92 × 10^–9^ $m^{2}\cdot s^{-1}$
$c_{{\mathrm{HCO}}_{3}^{-}}^{0}$	1 × 10^–6^ $\mathrm{mol}\cdot m^{-3}$	$D_{{\mathrm{HCO}}_{3}{}^{-}}$	1.19 × 10^–9^ $m^{2}\cdot s^{-1}$
$c_{{\mathrm{Zn}}^{2+}}^{0}$,${\;c}_{{\mathrm{ZnOH}}^{+}}^{0},$ $c_{\mathrm{Zn}{\left(\mathrm{OH}\right)}_{2}}^{0}$, $c_{{\mathrm{ZnCO}}_{3}}^{0}$	0	$D_{{\mathrm{Zn}}^{2+}}$,$D_{{\mathrm{ZnOH}}^{+}}$, $D_{\mathrm{Zn}{\left(\mathrm{OH}\right)}_{2}}$, $D_{\mathrm{Zn}{\left(\mathrm{CO}\right)}_{3}}$	0.7 × 10^–9^ $m^{2}\cdot s^{-1}$
$k_{O_{2}}$	2.5 × 10^–3^ $m^{2}\cdot s^{-1}$	$k_{\mathrm{Zn}}$	2 × 10^–8^ $\mathrm{mol}\cdot m^{2}\cdot s^{-1}$
$\alpha_{O_{2}}$	0.15	$\alpha_{\mathrm{Zn}}$	0.40

**Table 2 materials-15-03764-t002:** Equilibrium reactions considered in the simulation study and their equilibrium constants. The chemical reactions are homogeneous in the electrolyte [9,20,21].

Chemical Reaction	log(K) at 25 °C
$H^{+}+{\mathrm{OH}}^{-}=H_{2}O$	14
${\mathrm{Zn}}^{2+}+{\mathrm{OH}}^{-}={\mathrm{ZnOH}}^{+}$	5.04
${\mathrm{ZnOH}}^{+}+{\;\mathrm{OH}}^{-}=\mathrm{Zn}{\left(\mathrm{OH}\right)}_{2}$	6.06
${\mathrm{CO}}_{2}+{\mathrm{OH}}^{-}={\mathrm{HCO}}_{3}^{-}$	7.65
${\mathrm{HCO}}_{3}^{-}+{\mathrm{OH}}^{-}={\mathrm{CO}}_{3}^{2-}+H_{2}O$	3.67
${\mathrm{Zn}}^{2+}+{\mathrm{CO}}_{3}^{2-}={\mathrm{ZnCO}}_{3}$	5.3

## Data Availability

The data required to reproduce these findings cannot be shared at this time as they are an important part of our ongoing research.

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
