# Peer review of "Can Finite Element Method Obtain SVET Current Densities Closer to True Localized Corrosion Rates?"

_materials, 2022, doi:10.3390/ma15113764_

Round 1

Reviewer 1 Report

In this study, finite element method (FEM) was used to simulate the response of Scanning Vibrating Electrode Technique (SVET) across an iron–zinc cut-edge sample in order to provide a deeper understanding of localized corrosion rates measured by SVET. This study achieves good originality but is insufficient for acceptance in as-received form. I think that the paper should be improved in these aspects:

  1. The level of English should be further improved. For example, the sentence of “First, it necessary to briefly explained what is meant by “simple” models” should be changed into “First, it is necessary to briefly explained what is meant by “simple” models”. In addition, please check misprints, e.g. “Thébault et al. studied extensively studied the simulation of iron-zinc cut-edge corrosion”.
  2. In Introduction part, the advantages of SVET should be provided and therefore some information should be added. For example, “At present, various electrochemical methods have been employed to assess the anti-corrosion performance, such as potentiodynamic polarization measurements (Shuangshuang Chen, JianxinTu, Qiang Hu, Xinbo Xiong, Jiajia Wu, Jizhao Zou, Xierong Zeng, Corrosion resistance and in vitro bioactivity of Si-containing coating prepared on a biodegradable Mg-Zn-Ca bulk metallic glass by micro-arc oxidation [J], Journal of Non-Crystalline Solids, 456(2017)125-131.), electrochemical impedance spectroscopy (EIS) (Xiaoting Shi, Yu Wang, Hongyu Li, et al, Corrosion resistance and biocompatibility of calcium-containing coatings developed in near-neutral solutions containing phytic acid and phosphoric acid on AZ31B alloy, Journal of Alloys and Compounds, 2020, 823, 153721), electrochemical noise methods, et al. Compared with the electrochemical methods as above, SVET is a valuable technique and it has some advantages, for example, ….”.
  3. Because of the electrolyte diffusion by a vibrating SVET probe, the measured condition by SVET is not a true representation of galvanic corrosion. The authors should provide the error value between SVET current density and LECC-SVET current density.

Author Response

  1. We thank the reviewer for pinpointing the typos. A thorough English check was run throughout the manuscript.
  2. The suggested references are now studied and added.
  3. The "deviation" is now added.

Reviewer 2 Report

In this submission to Materials, the authors use finite element methods to simulate the response of Scanning Vibrating Electrode Techniques (SVET) across an iron–zinc cut-edge sample to provide a deeper understanding of localized corrosion rates measured by SVET. The authors find that if the diffusion layer is neglected, the simulated current density using the Laplace equation fits perfectly the experimental SVET current density. The authors also note that the electrolyte is not perturbed by a vibrating SVET probe, so a diffusion layer does exist. The authors conclude that although the LECC-SVET current density did not fit the experimental SVET current density as that obtained from the Laplace equation, it likely represents current densities closer to true, unperturbed, corrosion conditions than the SVET data from bulk conductivity.

I find this manuscript to be of interest to modelers for diffusion in electrolytes as well as readers of this journal. As such, I am somewhat in favor of publication with a few minor edits that should be incorporated in the next revision. In particular, there has been much prior work using atomistic-to-continuum simulations to understand electrolyte systems using both atomistic and finite element methods, which should be noted:

J. Chem. Theory Comput. 2011, 7, 1736–1749
Sci. Rep. 2019, 9, 16081

In particular, these prior works showed that atomistic combined with finite-element simulations can give additional mechanistic information at the atomic level for these interfaces, which should be mentioned. I am not asking the authors to carry out such atomistic simulations, but it should be mentioned that these types of computational approaches have been used in prior work. With these minor edits, I would be willing to re-review this manuscript for subsequent publication in Materials.

Author Response

We thank the reviewer's inputs. The suggested literature were studied and cited.

Reviewer 3 Report

In this article, the authors try to present high-precision methods for studying corrosion rates and compare their advantages and benefits. Finally to consider the accuracy of finite element simulations for this purpose, they use local electrolyte conductivity instead of bulk electrolyte conductivity measured experimentally and also by employing Nernst-Einstein equation. Eventually, the authors proposed a new algorithm as LECC-SVET.

The written introduction was very comprehensive and complete, which shows the authors' mastery of the subject under study. In other words, they are known as an expert in the field of corrosion. In addition,

The materials and methods section is described in great detail so that another researcher can repeat them step by step and achieve a similar specimen. This is a strength point of this article.

And the great advantage of the present work was the scientific interpretation of the results. Also, they proposed a new method to assess and check the corrosion rate.

In summary, I strongly suggested to publish the paper in the present form. 

Author Response

We greatly appreciate the reviewer's nice comments.